# Chronos: A Unified Framework for Predicting Training Time and Convergence in Deep Learning

## Abstract

Training deep neural networks requires extensive experimentation to estimate how long it will take, and how much it will cost, for a model to reach convergence. CHRONOS is the first end-to-end framework that predicts both the number of epochs to convergence and the total training time and cost before full training is performed for the target configuration. It integrates computational features that represent model behavior with early signal probes capturing initialization dynamics from a single mini-batch. Leveraging these signals, CHRONOS learns a cross-architecture mapping that generalizes across models and hardware configurations. It provides zero-shot estimates of convergence and cost for new target configurations without performing full training for those configurations. Across a diverse spectrum of architectures, from lightweight models such as MobileNetV2 and DeiT-Tiny to deeper and more complex networks, including ViT, ResNet-50, and DenseNet-121, CHRONOS achieves an average prediction error of 13.7% MAPE for iteration-level execution time and 22.1% MAPE for convergence estimation, demonstrating robust generalization across model scales and training complexities.

## 1 Introduction

Deep neural networks (DNNs) have driven major advances across vision and language (He et al., 2016; Huang et al., 2017; Krizhevsky et al., 2012; Simonyan & Zisserman, 2015; Vaswani et al., 2017), and today's large models push scale to billions of parameters (Devlin et al., 2019; Brown et al., 2020; Wang et al., 2023; Arfeen et al., 2025). These gains arrive with escalating computational demands and longer training times (Park et al., 2020; Liang et al., 2025; Ibrahim et al., 2024). Despite widespread adoption, training DNNs remains resource intensive (Jung et al., 2019; Coleman et al., 2017; Mattson et al., 2020). In practice, practitioners must choose hyperparameters (e.g., learning rate and batch size) and architectures within hardware constraints, often via expensive search procedures.

Imagine being a researcher with countless ideas but limited time and computational resources to explore them. You plan to launch thousands of experiments, yet as deadlines approach, you start to wonder not only whether they will finish on time and within budget, but also how long each experiment might take to train. *What if you could predict the total training time and the number of epochs required for a network to converge before running a single experiment?*

This gap underscores the need for a more general and predictive approach to performance estimation, one that can anticipate training behavior without relying on direct GPU execution or limited benchmark data. Existing benchmarks (e.g., MLPerf (Mattson et al., 2020)) rarely capture the diversity of custom DNN architectures used in research, leaving practitioners without reliable guidance when evaluating novel models on specific hardware.

Improving the efficiency of deep learning training requires accurate modeling of execution dynamics across architectures and hardware. Analytical models (Bai et al., 2022; Moolchandani et al., 2023) provide coarse estimates based on network structure, but overlook runtime factors such as memory access and parallel inefficiencies. Runtime-based approaches (Geoffrey et al., 2021; Lee et al., 2025) improve accuracy using GPU traces, but depend on profiling and known architectures. Learning-based methods (Justus et al., 2018;

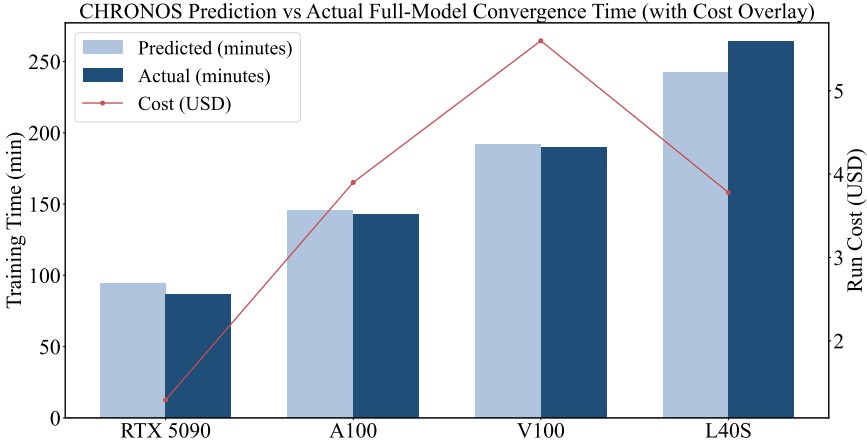

Figure 1: CHRONOS prediction versus actual full model convergence time for the Vision Transformer (ViT) (Dosovitskiy et al., 2020) on CIFAR-100 (Krizhevsky et al., 2009). This figure highlights how CHRONOS reveals cost–performance trade-offs across GPUs, showing that consumer-grade hardware like the RTX 5090 can approach or exceed the convergence speed of data-center GPUs at lower cost. Section 4 details the experimental setup.

Pourali et al., 2025) leverage computational features to generalize across hardware, yet they treat layers independently and ignore inter-layer dependencies like activation reuse and scheduling effects. Addressing these dependencies is key to achieving accurate, end-to-end predictions of training time and cost.

As discussed in Section 2, existing predictive frameworks still struggle to capture the complex dependencies that govern the training behavior of modern deep networks. As models increase in scale and architectural diversity, their performance becomes heavily influenced by cross-layer interactions, activation reuse, and hardware scheduling effects that traditional analytical or layer-wise models fail to represent. Moreover, most current methods treat convergence and runtime as separate problems, overlooking how optimization dynamics and computational characteristics jointly determine overall training efficiency.

```python
import chronos

predictor = chronos.TrainingTimePredictor(
    model=chronos.Models.ViT,
    dataset=chronos.Datasets.STL10,
    batch_size=16,
    learning_rate=0.001,
    optimizer=chronos.Optimizers.Adam,
    precision=16,
    gpu=chronos.Devices.V100,
)

# Predict per-epoch time and total number of epochs
epoch_time_s = predictor.predict_epoch_time()
num_epochs = predictor.predict_number_of_epochs()

# Compute total estimated training time in hours
total_time_hours = (epoch_time_s * num_epochs) / 3600

print(f"Each training epoch is predicted to take {epoch_time_s:.2f} seconds.")
print(f"The total number of epochs is predicted to be {num_epochs}.")
print(f"The total training time is predicted to be {total_time_hours:.2f} hours.")
```

Listing 1: An example of how CHRONOS can be used to predict both per epoch time and total training duration for a given model–dataset configuration.

To overcome these challenges, we propose **Chronos**, a unified framework that predicts end-to-end training time from model initialization by combining computational features with early training signals. CHRONOS efficiently

estimates the number of epochs to convergence and total training duration for a new target configuration without requiring that configuration to be fully trained, addressing the difficult and underexplored problem of inferring learning dynamics from initialization in nonlinear deep networks.

Figure 1 illustrates how CHRONOS models both convergence dynamics and cost efficiency across heterogeneous GPUs. In this experiment, a Vision Transformer (ViT) (Dosovitskiy et al., 2020) was trained on CIFAR-100 (Krizhevsky et al., 2009) using the AdamW optimizer with learning rate $5 \times 10^{-4}$, batch size 4, and 16-bit precision. CHRONOS accurately captures the convergence trajectory, demonstrating strong agreement between the predicted and observed training durations. Beyond prediction accuracy, CHRONOS reveals practical trade-offs between training time and monetary cost. High-end accelerators such as the A100 and V100 achieve faster convergence but at higher expense, while consumer grade GPUs like the RTX 5090 provide competitive performance at a fraction of the cost. Although the L40S is the cheapest to rent, it exhibits slower convergence. Users can still obtain faster overall training by opting for the A100, which, despite being roughly twice as costly per hour, completes runs significantly sooner. These results illustrate how CHRONOS can inform GPU selection by quantifying the balance between runtime efficiency and cost. The framework integrates seamlessly into existing workflows (see Listing 1) and is available as open source[1].

The main contributions of our work are the following:

(i) CHRONOS integrates computational feature modeling with initialization based probing to jointly predict iteration-level execution time and convergence behavior across diverse architectures and GPU platforms.

(ii) It leverages hardware aware structural features together with single batch signals such as gradient norm, NTK trace, and initial loss to provide zero-shot estimates of per-iteration cost and the number of epochs to convergence without performing full training.

(iii) Across a diverse spectrum of architectures and hardware tiers, CHRONOS achieves an average prediction error of 13.7% MAPE for iteration-level execution time and 22.1% MAPE for convergence estimation, demonstrating consistent generalization across unseen models and GPUs.

## 2    Related Work

This section provides an overview of prior work on predicting the performance of deep neural network (DNN) training. We first discuss existing approaches for estimating training time across architectures and hardware platforms, followed by methods for forecasting model convergence behavior. Together, these topics highlight the separate evolution of runtime and optimization modeling and motivate the need for a unified framework capable of jointly predicting end-to-end training time and cost.

### 2.1    Training Time Prediction

Predicting the training time of deep neural networks (DNNs) is essential for efficient resource allocation and system design. Early *analytical models* approached this problem by deriving execution cost from network topology and operation counts, typically using parameters, FLOPs, or layer depth as proxies for runtime (Bai et al., 2022; Moolchandani et al., 2023). Although these methods provide coarse grained estimates, they fail to account for runtime phenomena such as memory bandwidth constraints, data loading overheads, or GPU scheduling inefficiencies, leading to large deviations from actual measurements.

*Profiling-based methods* offer finer grained accuracy by utilizing traces collected from frameworks like TensorFlow and PyTorch or from standardized benchmarks such as MLPerf (Mattson et al., 2020). These systems, such as Habitat (Geoffrey et al., 2021) and Lumos (Liang et al., 2025), construct detailed execution graphs that model kernel level dependencies, allowing simulation or replay of training steps on new configurations. However, they require executing part of the model, making them unsuitable for pre-training estimation or large scale design exploration.

---

[1]github.com/anonymousgithacc/chronos

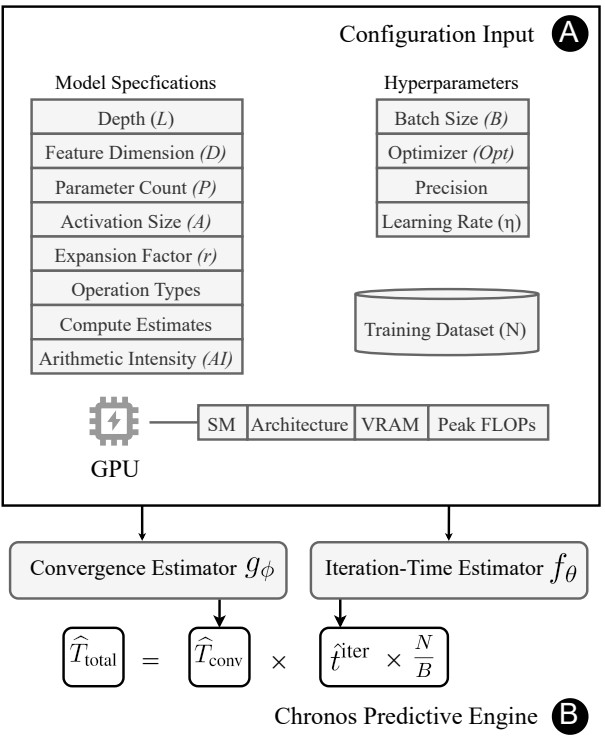

Figure 2: System overview of the CHRONOS framework. The configuration input (model, hyperparameters, dataset, and GPU) feeds into the predictive engine, which estimates both iteration time and convergence. Their combination yields the predicted total training time and cost.

More recently, *learning-based prediction frameworks* have emerged to bridge generality and efficiency. PreNeT (Pourali et al., 2025) introduced a data driven approach that learns iteration level execution time directly from computational features such as FLOPs, parameter bytes, activation sizes, and arithmetic intensity combined with GPU descriptors. These methods generalize across architectures and hardware, providing accurate runtime prediction without explicit profiling. Yet, despite their success, they remain limited to per iteration or per layer timing and cannot infer how long training will continue before convergence. CHRONOS builds on this line of work but targets a different prediction objective. While PRENET focuses on estimating training time from layer-level computational features, CHRONOS predicts end-to-end time-to-convergence by combining an iteration-time estimator with a convergence estimator. The model-level runtime representation improves the timing component by capturing cross-layer interactions, while the broader distinction is that CHRONOS combines runtime prediction with convergence-horizon estimation to predict end-to-end training time and cost. This allows CHRONOS to estimate total training duration and cost, which cannot be obtained from iteration-time prediction alone.

## 2.2 Convergence Prediction

Convergence prediction aims to estimate how many epochs or optimization steps are required for a model to reach a target loss or accuracy. Early approaches addressed this problem through *learning-curve extrapolation*, where partial validation curves are fitted to parametric functions in order to forecast final performance or determine early stopping decisions (Domhan et al., 2015; Klein et al., 2017). While useful for adaptive scheduling and hyperparameter search, these methods require observing a portion of the training trajectory and therefore do not provide a zero-shot estimate before training begins.

A second line of work studies convergence behavior using signals available at initialization or near the start of training. Neural Tangent Kernel (NTK) based analyses show that the structure of the gradient Gram matrix can characterize optimization dynamics in over-parameterized networks (Jacot et al., 2018). Related

approaches use gradient statistics, spectral properties, or probing-based quantities to estimate trainability and convergence behavior (Park et al., 2020; Lee et al., 2025). These methods provide important evidence that early optimization geometry is informative, but they typically focus on convergence behavior itself rather than end-to-end training time or cost.

A particularly relevant approach predicts the number of fine-tuning steps for a pre-trained network by linearizing the model around its initial weights, constructing the empirical gradient Gram matrix, and approximating the resulting optimization dynamics in function space (Zancato et al., 2020). This provides a principled connection between NTK structure and convergence speed, but its objective is to estimate fine-tuning steps for pre-trained models by solving an approximate training trajectory using the empirical NTK or a reduced representation. In contrast, CHRONOS does not simulate the training dynamics or solve a linearized stochastic or ordinary differential equation (SDE/ODE) for the target configuration. Instead, it learns a meta-level mapping from lightweight initialization-time probe features, static model and dataset descriptors, and training configuration variables to the convergence epoch.

Most prior work treats convergence estimation and runtime prediction as separate problems. This separation is reasonable when the goal is only to predict the number of epochs or optimization steps, since convergence is primarily governed by model architecture, dataset properties, initialization, and optimization settings such as learning rate, batch size, and optimizer choice. However, practical training planning requires more than the convergence horizon: users also need to know the wall-clock cost of each iteration on the target hardware. Thus, convergence prediction and runtime prediction become coupled at the level of end-to-end training time, where the total duration depends jointly on the number of epochs to convergence and the per-iteration execution time.

CHRONOS reflects this separation explicitly. Its convergence estimator uses initialization-time probe features and training configuration descriptors to estimate the convergence horizon, while its iteration-time estimator uses computational and hardware-aware features to estimate per-iteration runtime. The two estimates are then combined to predict total training time and monetary cost. This distinguishes CHRONOS from convergence-only methods and runtime-only frameworks such as PRENET (Pourali et al., 2025) and HABITAT (Geoffrey et al., 2021).

## 3 Chronos Design

Figure 2 provides an overview of the CHRONOS framework. As shown in **Ⓐ**, the framework begins with the *Configuration Input* stage, which automatically extracts all static descriptors defining the model architecture, hyperparameters, dataset properties, and GPU hardware configuration. These descriptors constitute the analytical foundation for the predictive modules that follow. To ensure generality across architectures, CHRONOS represents each model through a set of analytical *model specification components* summarized in Table 1. These parameters capture the structural and computational characteristics of the network and are used to derive higher level analytical features such as FLOPs, memory footprint, and arithmetic intensity.

In **Ⓑ**, the *CHRONOS Predictive Engine* integrates two complementary modules: the *iteration time estimator* (Section 3.1), which models the cost of executing one training iteration, and the *convergence estimator* (Section 3.4), which predicts how many epochs are required for the model to reach stability. These two estimators answer different questions. The iteration time estimator determines the cost of a single optimization step on a given GPU, while the convergence estimator determines how many such steps are expected before convergence. Thus, measuring or predicting a single iteration time alone is insufficient for estimating end-to-end training duration, because it does not reveal how many epochs the model will need to converge. The final total time calculation combines these two quantities with the number of mini batches per epoch, as formalized later in Section 3.4. Both estimators are trained offline and remain fixed during prediction for a new configuration. Therefore, the zero-shot prediction process uses the learned mappings only for inference; it does not update $f_\theta$ or $g_\phi$ online.

Table 1: Analytical model specification components used by CHRONOS.

| Component | Description |
|---|---|
| **Depth** ($L$) | Total number of parameterized layers or blocks in the model; controls overall network complexity. |
| **Feature Dimension** ($D$) | Width or embedding size of each layer; determines the size of intermediate feature representations. |
| **Parameter Count** ($P$) | Total number of trainable weights in the model; used to compute parameter and optimizer memory. |
| **Activation Size** ($A$) | Estimated number of activations produced during the forward pass; determines activation memory usage. |
| **Expansion Factor** ($r$) | Ratio controlling hidden layer widening in MLP or feed-forward blocks, such as token-mixing or feed-forward network expansions. |
| **Operation Types** | Distinguishes dominant layer types, such as convolution, linear, attention, and normalization, used to derive analytical FLOPs. |
| **Compute Estimates** | Aggregated analytical FLOPs across all layers, representing the total computational cost per iteration. |
| **Arithmetic Intensity** ($AI$) | Ratio of compute to memory operations, FLOPs/Bytes, reflecting the balance between computation and data movement. |

## 3.1 Iteration Time Estimator

The iteration time prediction module in CHRONOS estimates the average compute duration of a single training iteration, comprising forward, backward, and optimizer update operations across diverse neural network architectures and GPU hardware. Instead of relying on extensive runtime profiling, CHRONOS derives analytical features (Section 3.2) that capture the computational and memory characteristics of each model configuration, allowing accurate prediction without executing full training runs. This design is inspired by the analytical modeling approach introduced in PRENET (Pourali et al., 2025), which predicts per epoch training time using layer level computational features.

However, unlike PRENET (Pourali et al., 2025), which treats layers independently and aggregates their costs additively, CHRONOS models the inter-layer computational dependencies and accounts for the effects of optimizer states, numerical precision, and hardware specific execution characteristics. As a result, CHRONOS operates at finer granularity, predicting iteration level compute time while remaining portable across architectures and GPU types. Formally, for a configuration $c$ defined by its model architecture, dataset, batch size, optimizer, precision, and GPU type, the compute time per iteration is expressed as

$$t^{\text{iter}}(c) = f_\theta\big(\mathbf{x}(c)\big), \tag{1}$$

where $\mathbf{x}(c)$ denotes the feature vector encoding architecture and hardware dependent descriptors, and $f_\theta(\cdot)$ represents the learned regression function. The feature set $\mathbf{x}(c)$ includes analytical estimations of floating point operations (FLOPs), parameter and activation memory, optimizer state size, arithmetic intensity, and precision specific compute efficiency.

By modeling the iteration level compute cost rather than the full training duration, CHRONOS achieves fine grained timing prediction that generalizes across convolutional, token-mixing MLP, and transformer based architectures, while maintaining portability across GPUs, batch sizes, and numerical precisions.

## 3.2 Feature Formulation

To enable accurate iteration time prediction across heterogeneous model architectures and hardware configurations, CHRONOS constructs a unified analytical feature vector $\mathbf{x}(c)$ for each configuration $c$. This vector captures both computational and memory descriptors derived analytically from model topology and GPU specifications, ensuring portability without empirical profiling.

Table 2: Analytical feature vector $\mathbf{x}(c)$ used by CHRONOS.

| Component | Description |
|---|---|
| FLOPS$_{\text{train}}$ | Floating-point operations per training iteration, including both forward and backward passes. |
| Param$_{\text{bytes}}$ | Memory size of learnable parameters. |
| Act$_{\text{bytes}}$ | Estimated activation memory during the forward pass. |
| Opt$_{\text{bytes}}$ | Memory required by optimizer states. |
| AI | Arithmetic intensity, defined as the compute-to-memory ratio. |
| Batch | Mini-batch size per training iteration. |
| Precision | Numerical precision used during training, such as FP16 or FP32. |
| GPU$_{\text{specs}}$ | GPU descriptors, including SM count, peak FLOPs, and VRAM. |

Formally, a configuration is represented as

$$\mathbf{x}(c) = [x_1, x_2, \ldots, x_8],$$

where each element corresponds to a feature summarized in Table 2. Together, these features characterize the theoretical compute, memory, and hardware properties governing iteration level training time.

**Training FLOPs.** The total number of floating point operations required to complete a single training iteration is estimated as

$$\text{FLOPs}_{\text{train}} = \kappa \cdot \text{FLOPs}_{\text{fwd}}, \tag{2}$$

where $\text{FLOPs}_{\text{fwd}}$ represents the analytical count of operations in the forward pass, and $\kappa$ is a scaling factor that accounts for additional computation in the backward pass and optimizer updates. In practice, backward propagation is commonly approximated as requiring about twice the compute of the forward pass, while optimizer-specific operations, such as gradient accumulation and parameter updates in *AdamW*, introduce additional but comparatively smaller overhead. Accordingly, we model the training-FLOPs multiplier $\kappa$ as an architecture-aware bounded factor that typically falls between 2.5 and 3.5. Values near 2.5 are appropriate for simpler feed-forward or convolutional architectures, where the backward computation closely follows the forward graph. Values closer to 3.5 better reflect architectures with attention, normalization, or multi-branch structures, where backward dependencies and optimizer-state operations introduce greater computational overhead.

This bounded range is not tuned per test instance; rather, it provides a practical analytical approximation of full training compute cost without requiring explicit gradient tracing. The computation of $\text{FLOPs}_{\text{fwd}}$ differs across model families and is derived analytically from their layer topologies, as detailed below for convolutional, token-mixing, and transformer-based architectures.

**Convolutional Networks (CNNs).** For a convolution layer with input channels $C_{\text{in}}$, output channels $C_{\text{out}}$, kernel size $(K_h, K_w)$, and output feature map size $(H_{\text{out}}, W_{\text{out}})$ using $g$ groups:

$$\text{FLOPs}_\ell^{\text{conv}} = 2 \times \frac{C_{\text{in}}}{g} \times C_{\text{out}} \times K_h \times K_w \times H_{\text{out}} \times W_{\text{out}} \tag{3}$$

Fully connected layers contribute:

$$\text{FLOPs}_\ell^{\text{fc}} = 2 \times d_{\text{in}} \times d_{\text{out}}, \tag{4}$$

where $d_{\text{in}}$ and $d_{\text{out}}$ represent the number of input and output units in the layer, respectively, and the total forward FLOPs are obtained by summing across all layers:

$$\text{FLOPs}_{\text{fwd}}^{\text{CNN}} = \sum_{\ell \in \text{Conv}} \text{FLOPs}_\ell^{\text{conv}} + \sum_{\ell \in \text{FC}} \text{FLOPs}_\ell^{\text{fc}} \tag{5}$$

**Token-Mixing MLPs (MLP-Mixer, ResMLP, AS-MLP).** Let the image be divided into $P$ patches (tokens) of embedding dimension $D$, with each block containing token-mixing and channel-mixing MLPs of

expansion ratios $r_t$ and $r_c$, respectively. For $L$ stacked blocks,

$$\text{FLOPs}_{\text{fwd}}^{\text{MLP}} = L \times 4PD(r_t + r_c) \tag{6}$$

Variants such as ResMLP and AS-MLP modify the mixing operation (affine or axial), but maintain the same analytical structure.

**Transformer Architectures (DeiT, ViT).** For sequence length $T$, embedding dimension $D$, number of heads $H$ (with per head size $d_h = D/H$), feed-forward width $d_{\text{ff}}$, and $L$ transformer blocks,

$$\text{FLOPs}_{\text{fwd}}^{\text{Tr}} = L \times \left(4TD^2 + 2HT^2 d_h + 4TD d_{\text{ff}}\right), \tag{7}$$

where the first two terms correspond to the multi head self-attention mechanism, and the last term accounts for the feed-forward network.

**Memory Footprint.** To represent the total memory requirement per iteration, CHRONOS aggregates the parameter, optimizer, and activation storage:

$$\text{Mem}_{\text{total}} = \text{Param}_{\text{bytes}} + \text{Opt}_{\text{bytes}} + \text{Act}_{\text{bytes}}, \tag{8}$$

where $\text{Opt}_{\text{bytes}} = m_{\text{opt}} \cdot \text{Param}_{\text{bytes}}$ and $m_{\text{opt}} \in \{1, 2\}$ depending on whether the optimizer maintains additional states (e.g., $m_{\text{opt}} = 2$ for Adam/AdamW).

**Arithmetic Intensity.** Finally, CHRONOS introduces a roofline style measure of compute-to-memory balance:

$$\text{AI} = \frac{\text{FLOPs}_{\text{train}}}{\text{Mem}_{\text{total}}}, \tag{9}$$

which normalizes compute efficiency across GPUs with differing peak throughput and memory bandwidth.

This unified analytical formulation enables CHRONOS to describe convolutional, token-mixing, and transformer-based models within a shared feature space, providing a portable foundation for accurate iteration-time prediction across heterogeneous hardware.

### 3.3 Regression and Prediction Mapping

After constructing the analytical feature representation, CHRONOS learns the mapping function $f_\theta$ that predicts the iteration-level compute time from these features. This mapping is trained offline, before deployment, using a benchmark corpus of model–dataset–hardware configurations for which ground truth iteration times have been measured. During this offline stage, each configuration $c$ is converted into its analytical feature vector $\mathbf{x}(c)$, and $f_\theta$ is optimized to minimize the prediction error between the estimated iteration time $\hat{t}^{\text{iter}}$ and the measured iteration time $t^{\text{iter}}$:

$$\min_{\theta} \ \mathbb{E}_{c \sim \mathcal{D}}\left[\left(t^{\text{iter}}(c) - f_\theta(\mathbf{x}(c))\right)^2\right], \tag{10}$$

where $\mathcal{D}$ denotes the set of configurations across model architectures, datasets, and GPU hardware. Once trained, $f_\theta$ is kept fixed during prediction. For a new, unseen configuration, CHRONOS extracts $\mathbf{x}(c)$ from the model architecture, dataset, training configuration, and GPU descriptors, and then applies the trained regressor to obtain $\hat{t}^{\text{iter}}$. No online updates to $f_\theta$ are performed during this inference step. If new hardware platforms, model families, or training regimes are added later, $f_\theta$ can be periodically retrained or fine-tuned offline using the expanded benchmark corpus.

In practice, CHRONOS employs non parametric regressors such as Gradient Boosted Trees to capture the nonlinear relationships between computational features and iteration time. This approach provides strong interpretability and generalization across unseen models and GPUs, while maintaining low training cost. Compared to the per layer linear regression approach used in PRENET (Pourali et al., 2025), this formulation enables CHRONOS to jointly learn cross-layer dependencies and hardware-specific effects through feature interactions within $f_\theta$.

The resulting predictor produces an accurate estimate of iteration-level compute time, denoted as

$$\hat{t}^{\text{iter}} = f_\theta(\mathbf{x}(c)), \tag{11}$$

which can be aggregated over the predicted number of convergence epochs to compute the total training duration in the convergence module introduced next.

### 3.4 Convergence Estimator

While iteration time prediction captures the compute cost of a single optimization step, the total training duration of a model also depends on the number of iterations (or epochs) required for convergence. To estimate this component, CHRONOS introduces a convergence prediction stage that infers the expected number of training epochs needed for a model to reach a stable validation accuracy. This stage complements the iteration time predictor by modeling the learning dynamics that govern convergence behavior across architectures and optimization regimes.

Formally, for a configuration $c$ defined by its model architecture, dataset, learning rate, and optimizer, the convergence target $T_{\text{conv}}(c)$ represents the number of epochs required for the model to converge according to a validation based early stopping criterion. The goal of this stage is to learn a predictive function $g_\phi$ such that

$$T_{\text{conv}}(c) = g_\phi\big(\mathbf{z}(c)\big), \tag{12}$$

where $\mathbf{z}(c)$ denotes a convergence specific feature vector derived from analytical and statistical properties of the model and training configuration.

The feature vector $\mathbf{z}(c)$ includes both static and dynamic descriptors such as parameter count ($P$), dataset size ($N$), learning rate ($\eta$), batch size ($B$), and three probe-based signals, the gradient norm ($g^2$), NTK-trace proxy ($\tau$), and initial loss ($\ell_0$). Importantly, this probe is performed once at initialization. It consists of a single forward pass to compute the initial loss and a single backward pass to obtain gradients for $g^2$ and the NTK-trace estimate. No optimizer step is taken, and model parameters remain unchanged after probing. Therefore, the convergence estimator does not observe any training trajectory or validation curve.

**Gradient Norm Statistics.**  The gradient norm is computed once at model initialization, before any optimizer update is performed. Given a fixed probe mini-batch, model parameters $\theta_0$, and loss function $\mathcal{L}$, CHRONOS performs a single forward and backward pass and computes

$$g^2 = \frac{1}{|\theta|} \sum_{i=1}^{|\theta|} \|\nabla_{\theta_i}\mathcal{L}(\theta_0)\|_2^2, \tag{13}$$

where $|\theta|$ is the number of trainable parameters. No optimizer step is taken after this backward pass, and the model parameters remain unchanged. Thus, $g^2$ represents an initialization-time probe feature rather than a statistic monitored throughout training. It captures the magnitude of the initial update signal and provides a zero-shot indicator of early trainability and convergence behavior across architectures.

**Neural Tangent Kernel Trace.**  The Neural Tangent Kernel (NTK) quantifies how small parameter perturbations influence model outputs, and its trace ($\tau$) captures the aggregate scale of this sensitivity. For a model $f_\theta(x)$ parameterized by $\theta$, the empirical NTK over a set of samples $\mathcal{S}$ is defined as

$$K_{ij} = \nabla_\theta f_\theta(x_i)^\top \nabla_\theta f_\theta(x_j), \qquad x_i, x_j \in \mathcal{S}, \tag{14}$$

where $\nabla_\theta f_\theta(x_i)$ denotes the gradient of the model output with respect to its parameters. The normalized trace of this matrix is

$$\tau(\mathcal{S}) = \frac{1}{|\mathcal{S}|} \text{Tr}(K), \tag{15}$$

where $\text{Tr}(\cdot)$ denotes the matrix trace.

In CHRONOS, $\mathcal{S}$ is not the full training set. Instead, $\mathcal{S}$ is the fixed probe mini-batch used for the initialization-time probe. Thus, the NTK-based feature is computed once at initialization on a single mini batch and is not recomputed over the full dataset or across training epochs.

**Zero-Shot Probing.** A key distinction of CHRONOS is that convergence prediction operates in a *zero-shot* manner at prediction time. Instead of executing a partial training trajectory for the target model, CHRONOS performs a single initialization-time probe on one fixed mini-batch. The probe requires one forward pass and one backward pass to extract the initial loss $\ell_0$, gradient norm $g^2$, and NTK-trace proxy $\tau$. No optimizer update is performed during this step, and the model parameters remain unchanged after the probe. Thus, the estimator observes only initialization statistics and does not use partial training curves, validation feedback, or any post update model state from the target run.

The convergence estimator $g_\phi$ is trained offline using a meta-dataset of completed training runs, where each configuration is labeled with its validation-based convergence epoch $T_{\text{conv}}$. Therefore, CHRONOS does not infer convergence by extrapolating the first few iterations of the target model. Rather, it learns a cross-configuration mapping from initialization time descriptors and static training features to final convergence behavior observed in the offline corpus. This design drastically reduces prediction time overhead, since estimating convergence for a new configuration requires only the one time initialization probe rather than full or partial training.

Ground truth convergence labels are obtained through validation-based early stopping, where convergence is identified once the validation loss fails to improve by a margin $\delta$ for $p$ consecutive epochs. The predictor $g_\phi$ is trained to minimize the squared error between the predicted and actual convergence epochs:

$$\min_\phi \ \mathbb{E}_{c\sim\mathcal{D}}\big[\big(T_{\text{conv}}(c) - g_\phi(\mathbf{z}(c))\big)^2\big], \tag{16}$$

where $\mathcal{D}$ denotes the set of observed configurations.

The predicted convergence horizon $\widehat{T}_{\text{conv}}(c)$, expressed in epochs, is then combined with the predicted iteration cost to estimate total training time. For a dataset with $N$ training samples and batch size $B$, the total training duration is computed as

$$\widehat{T}_{\text{total}}(c) = \widehat{T}_{\text{conv}}(c) \times \frac{N}{B} \times \hat{t}^{\text{iter}}(c), \tag{17}$$

where $\hat{t}^{\text{iter}}(c)$ denotes the predicted time per training iteration.

## 4 Experiments

We first outline the experimental setup, including testbed and preprocessing details (Section 4.1), followed by evaluation metrics and baselines (Sections 4.2–4.3). We then examine generalization to unseen GPUs (Section 4.4), cross-model and optimizer sensitivity (Sections 4.5, 4.6), and conclude with an end-to-end ResNet-50 case study demonstrating full training-time prediction across various GPUs (Section 4.7).

### 4.1 Experiment Setup

**Testbed.** We evaluate CHRONOS across heterogeneous GPU platforms spanning both cloud and on premise environments. The experiments are conducted on GPUs provisioned from Google Cloud (L4, T4, Tesla V100, Tesla P4, Tesla P100) and RunPod servers (H100 PCIe, A100 80 GB PCIe, L40S, RTX 5090, A40). All nodes run CUDA 12.3 and cuDNN 9.0 with PyTorch 2.1. Each configuration is profiled per GPU with five warm up and twenty measured batches. The reported results represent the mean of all measured batches, ensuring stable estimates of per batch compute time.

**Benchmarks.** We benchmark a diverse set of deep neural architectures covering multiple families of modern networks: AsMLP (Lian et al., 2021), MLP-Mixer (Tolstikhin et al., 2021), ResMLP (Touvron et al., 2022), DenseNet-121 (Huang et al., 2017), MobileNetV2 (Sandler et al., 2018), ResNet-50 (He et al., 2016), VGG-16 (Simonyan & Zisserman, 2015), DeiT-Tiny (Touvron et al., 2021), and ViT (Dosovitskiy et al., 2020). These models are evaluated on widely used datasets, including CIFAR-10, CIFAR-100 (Krizhevsky et al., 2009), STL-10 (Coates et al., 2011), and TinyImageNet (Deng et al., 2009), covering both low and high resolution image domains. Each model–dataset pairing is trained under multiple hyperparameter configurations to populate a rich meta-dataset for prediction.

**Hyperparameter Sweeps.** CHRONOS systematically varies three key hyperparameters: learning rate $\in \{5 \times 10^{-4}, 1 \times 10^{-3}, 2 \times 10^{-3}\}$, batch size $\in \{4, 8, 16, 32, 48, 64\}$, and optimizer $\in \{\text{AdamW, Adam, SGD}\}$. These ranges align with standard practices established across recent deep learning studies. For vision models such as CNNs and MLP based architectures, similar learning rate magnitudes ($10^{-4}$ to $10^{-3}$) and batch sizes (32–256) are commonly adopted to balance training stability and throughput (He et al., 2016; Touvron et al., 2021; Tolstikhin et al., 2021). For transformer based models, the same order of magnitude for learning rates and small to medium batch sizes are consistent with fine tuning regimes reported in BERT and DeiT studies (Devlin et al., 2019; Sanh et al., 2019; Touvron et al., 2021). We include both adaptive (Adam/AdamW) and momentum based (SGD) optimizers, reflecting their respective prevalence in transformer and convolutional model families (Kingma & Ba, 2015; Loshchilov & Hutter, 2019; He et al., 2016). By sweeping across these canonical yet widely used hyperparameter ranges, CHRONOS captures how variations in optimization dynamics influence per batch compute time and convergence trends across heterogeneous architectures.

**Preprocessing and Feature Construction.** Before training, CHRONOS standardizes all numerical inputs and applies log scaling to highly skewed quantities to stabilize learning across GPUs. The feature space integrates hardware descriptors (e.g., architecture family, memory type, tensor core support) and batch related interactions that capture utilization and headroom behavior across devices. Variables directly correlated with runtime measurements are excluded to avoid leakage and ensure true cross-hardware generalization.

**Timing Trials.** For each configuration, CHRONOS executes controlled timing trials consisting of five warm up batches followed by twenty measured batches. All per batch metrics are synchronized using CUDA events. Each configuration is profiled once per GPU, and the reported values correspond to the mean over all measured batches to ensure stable estimates of per batch compute time.

**Convergence Tracking.** To estimate convergence duration, each configuration is trained with an 80/20 train–validation split using patience-based early stopping. The epoch at which validation loss ceases to improve is recorded as $T_{\text{conv}}$, representing the effective convergence horizon for that setting. Constructing the iteration-time and convergence meta datasets required approximately 480 GPU hours across cloud and on-premises servers to execute full training and timing sweeps for all model–dataset configurations. This cost is incurred once during offline meta-dataset construction and is amortized over future prediction queries, where the trained regressors are reused for new model–dataset–GPU configurations.

**Prediction Time Cost.** Once the regressors are trained, CHRONOS does not require full training for a new target configuration. The iteration-time estimate is obtained from analytical model and hardware features, while the convergence estimate requires only a single initialization-time mini-batch probe with one forward and backward pass and no optimizer update. For a dataset with $N$ training samples and batch size $B$, one full epoch requires approximately $\lceil N/B \rceil$ mini-batch iterations, whereas the convergence probe requires only one mini-batch. For example, with $N = 50{,}000$ and $B = 128$, one epoch requires approximately 391 mini-batch iterations, while the CHRONOS probe uses a single mini-batch of 128 samples. Thus, the probe is about 1/391, or less than 0.3%, of a single epoch before accounting for the omitted optimizer update.

## 4.2 Evaluation Metrics

To assess the predictive accuracy of CHRONOS, we employ the same evaluation metrics used in PRENET (Pourali et al., 2025): the Mean Absolute Percentage Error (MAPE) and the Root Mean Squared Error (RMSE).

$$\text{MAPE} = \frac{100}{N} \sum_{i=1}^{N} \left| \frac{y_i - \hat{y}_i}{y_i} \right| \tag{18}$$

$$\text{RMSE} = \sqrt{\frac{1}{N} \sum_{i=1}^{N} (y_i - \hat{y}_i)^2} \tag{19}$$

Table 3: Comparison of RMSE values between PRENET (Pourali et al., 2025) and CHRONOS across models. Chronos consistently reduces the prediction error, achieving up to 60% improvement over PreNeT.

| Model | PreNeT (RMSE) | Chronos (RMSE) | Improvement | Avg. Batch Time (ms) |
|---|---|---|---|---|
| ASMLP | 23.32 | 16.91 | 27% | 109.68 |
| MLP-Mixer | 34.54 | 13.86 | 60% | 106.97 |
| ResMLP | 25.43 | 11.53 | 55% | 73.30 |
| DenseNet-121 | 54.34 | 23.15 | 57% | 187.41 |
| MobileNetV2 | 25.43 | 11.40 | 55% | 58.32 |
| ResNet-50 | 43.45 | 22.95 | 47% | 189.95 |
| VGG-16 | 19.34 | 10.45 | 46% | 187.66 |
| DeiT-Tiny | 30.43 | 12.23 | 60% | 127.87 |
| ViT | 69.43 | 28.96 | 58% | 155.70 |

MAPE quantifies the relative deviation between predicted and actual values, offering an interpretable measure of percentage error, while RMSE reflects the overall magnitude of prediction deviations and penalizes larger discrepancies more strongly. Together, these metrics capture both precision and robustness across architectures and hardware configurations.

## 4.3 Baselines and Comparison Methodology

**Baselines.** For timing prediction, we compare CHRONOS against PRENET (Pourali et al., 2025), the most closely related framework for estimating training time. PRENET models the execution cost of deep networks by analyzing individual layers, where each layer's runtime is predicted from computational features such as FLOPs, parameter size, and memory utilization, and the total training time is obtained through aggregation. In contrast, CHRONOS adopts a holistic view that represents the model as a whole, capturing inter-layer dependencies, architectural structure, and hardware characteristics within a unified predictive space. By incorporating hardware specific ratios such as FLOPs-to-bandwidth and arithmetic-intensity-to-FLOPs, CHRONOS achieves more accurate scaling behavior across heterogeneous GPUs and improves generalization to unseen hardware configurations.

**Evaluation Protocol.** We conduct three complementary evaluations for timing prediction: (*i*) an 80/10/10 random split to assess in distribution accuracy, (*ii*) Leave-One-Dataset-Out (LODO) to test generalization across datasets, and (*iii*) Leave-One-Model-Out (LOMO) to evaluate cross-architecture robustness. Each setting is repeated for all CNN, MLP, and Transformer families. The regression model used in all evaluations is an XGBoost (Chen & Guestrin, 2016) regressor, selected based on preliminary evaluations. A detailed ablation study supporting this choice is available in our GitHub repository.

**Convergence Evaluation.** To the best of our knowledge, no prior work has attempted to predict the full training convergence horizon of deep neural networks directly from computational and probe-based features. Therefore, we evaluate CHRONOS independently for this component, focusing on its ability to predict $T_{\text{conv}}$, which is the epoch at which validation loss stops improving under early-stopping criteria. The same training and validation splits are used for all architectures to maintain consistency.

**Comparison Results.** Table 3 compares RMSE values between PRENET (Pourali et al., 2025) and CHRONOS using the unified evaluation protocol in Section 4.2. All models were trained under identical batch size and GPU configurations, with RMSE measured on the 80/20 train–test split. Average batch time (ms) is reported to indicate the computational scale of each model. CHRONOS consistently achieves lower prediction error, improving over PRENET by 27–60% across architectures, demonstrating stronger modeling of compute memory interactions and robustness across hardware tiers.

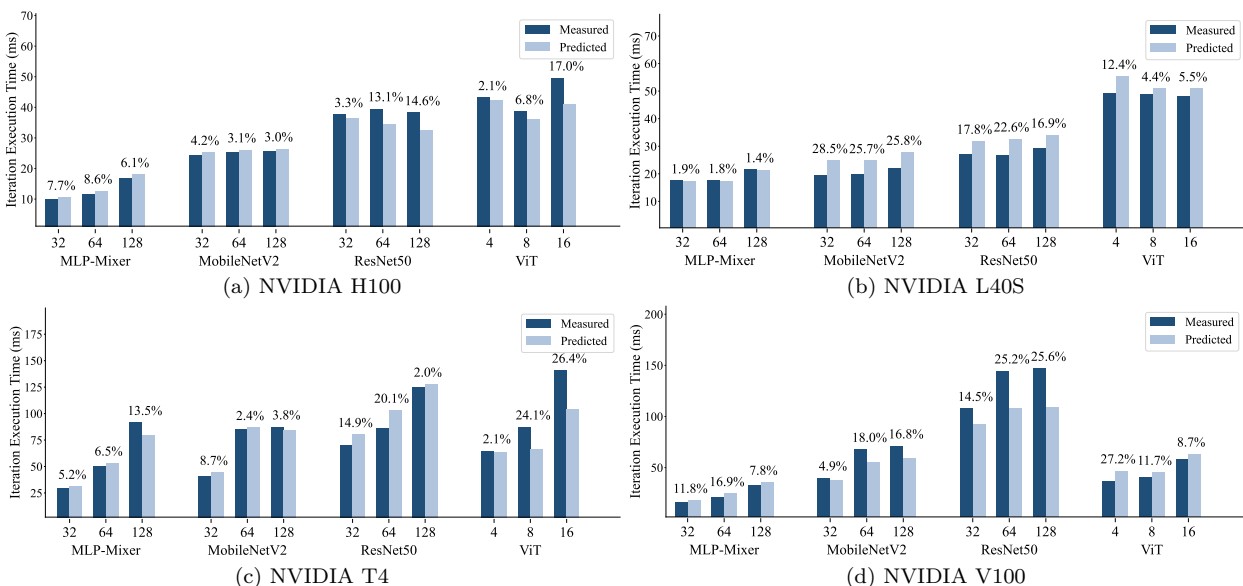

Figure 3: Measured vs. predicted iteration execution times for unseen GPUs across multiple batch size configurations. CHRONOS demonstrates strong cross-GPU generalization without retraining, accurately capturing batch size scaling trends across diverse architectures. All models, including MLP-Mixer, MobileNetV2, ResNet-50, and ViT, were trained using the Adam optimizer with a learning rate of 0.001. The percentage error is shown above each prediction.

## 4.4 Unseen GPU Evaluation

We evaluate CHRONOS using a leave-one-GPU-out protocol to assess its ability to generalize to held-out GPU configurations. For each experiment, the timing regressor is trained on measurements from all GPUs except the target GPU, and the held-out GPU is used only for evaluation. Thus, no timing samples from the target GPU are included during training.

The predictor operates on standardized and log-scaled numerical features, together with encoded descriptors capturing GPU architecture family, precision mode, memory technology, and hardware capacity. This preprocessing provides consistent scaling across GPU configurations and avoids leakage from timing-dependent quantities.

Figure 3 (a)–(d) presents representative evaluations for four held-out GPUs, showing measured versus predicted iteration times across multiple models and batch sizes. Across these devices, CHRONOS maintains close alignment between predicted and observed performance, indicating that the learned mapping can generalize across held-out GPU variants without using target-GPU timing samples.

This evaluation measures cross-GPU generalization within the evaluated hardware pool. Although some GPUs may share architectural components or belong to related architecture families, the held-out devices still differ in memory capacity, bandwidth, core counts, and deployment tier. The leave-one-GPU-out protocol therefore tests whether CHRONOS can predict performance on a target GPU without using timing samples from that GPU during training.

## 4.5 Cross-Model Generalization under LOMO Evaluation

To evaluate the generalization capability of CHRONOS across different architectures and GPU hardware, we conducted a *Leave-One-Model-Out (LOMO)* evaluation using our timing meta-datasets. In this setup, each model (e.g., VGG-16, ResNet-50, DenseNet-121, ViT) is excluded from training and reserved for testing, allowing us to assess the predictor's ability to estimate unseen architectures. This setup isolates model dependent factors while preserving cross GPU diversity. Figure 4 reports the mean absolute percentage

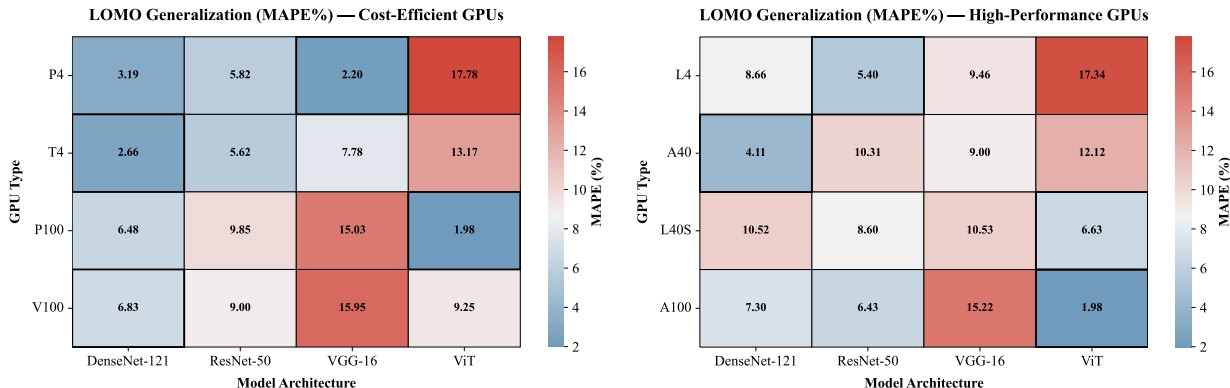

Figure 4: Cross model generalization results under LOMO evaluation, comparing prediction error (MAPE%) across GPU categories.

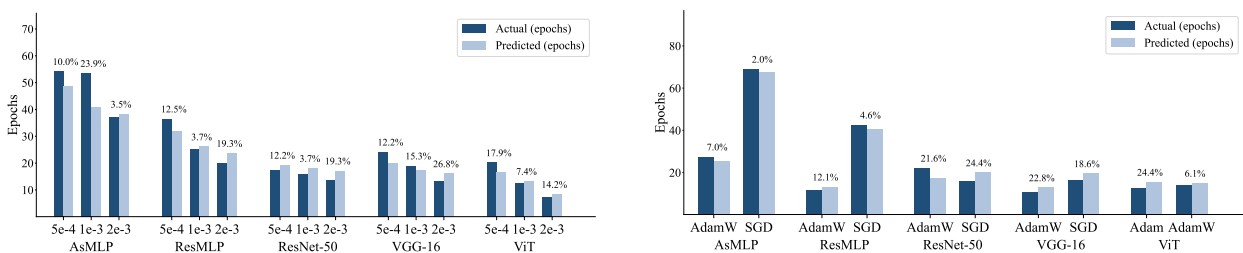

Figure 5: Comparison between measured and predicted convergence epochs across various learning rate and optimizer settings.

error (MAPE) across multiple GPUs, grouped into cost efficient and high-performance tiers. CHRONOS achieves consistently low prediction errors across both lightweight and compute-intensive models, with MAPE remaining below 15% for most configurations. Notably, prediction variance across GPUs such as the L40S, A100, and V100 remains minimal, indicating that the framework effectively adapts to diverse compute capabilities without overfitting to specific hardware. These results demonstrate strong cross-architecture and cross-hardware generalization in training time and convergence prediction.

### 4.6 Learning Rate and Optimizer Sweeps

We further evaluate the sensitivity of CHRONOS to optimization hyperparameters under a Leave-One-Dataset-Out (LODO) evaluation protocol. The purpose of this experiment is to test whether the convergence estimator can preserve learning-rate and optimizer-driven convergence trends when the target dataset is not observed during training. In each LODO run, one dataset is held out for testing, while the predictor is trained on the remaining datasets. We repeat this process across TinyImageNet, CIFAR-10, CIFAR-100, and STL-10.

The experiments cover AsMLP, ResMLP, ResNet-50, VGG-16, and ViT architectures under different learning rates ($5 \times 10^{-4}, 10^{-3}, 2 \times 10^{-3}$) and optimizer choices (AdamW, SGD, and Adam). Each model–dataset–hyperparameter configuration is evaluated independently rather than averaged across runs. Figure 5 reports measured and predicted convergence epochs for the held-out dataset cases. Across these configurations, CHRONOS captures the main empirical convergence trends, including faster convergence at larger learning rates in stable regimes and the improved stability of adaptive optimizers such as AdamW across diverse training conditions.

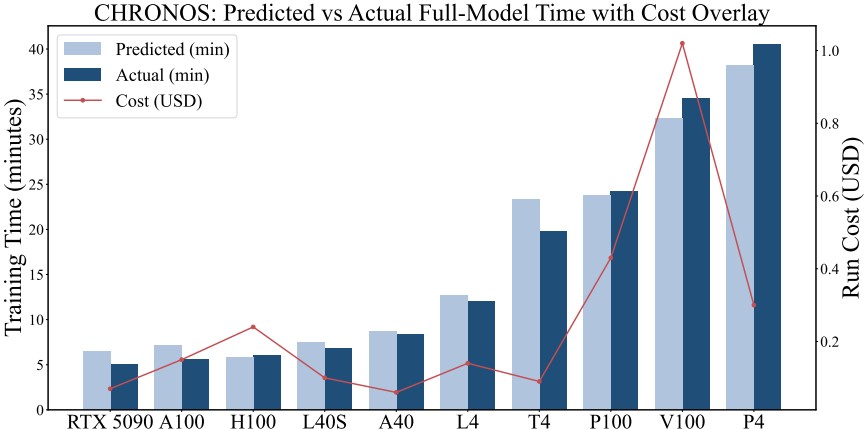

Figure 6: Predicted versus actual full model training time and cost for CHRONOS on ResNet-50 trained on TinyImageNet. Bars show predicted and measured training durations across GPU types, while the red line indicates run cost. The close alignment illustrates CHRONOS' ability to model end-to-end training time and cost across heterogeneous hardware.

### 4.7 End-to-End Prediction on ResNet-50

Figure 6 illustrates the end-to-end training-time prediction of a ResNet-50 model on the TinyImageNet dataset using a batch size of 128, learning rate $2.0 \times 10^{-3}$, and the SGD optimizer. CHRONOS achieves a prediction error of only 9.4% for this experiment, accurately modeling the relationship between hardware cost and convergence time across heterogeneous GPUs. Leveraging architectural and hardware-specific computational features, CHRONOS enables users to identify the most cost-efficient configuration for full training convergence. In this setup, the A40 emerges as the most budget-friendly GPU that completes training in a relatively short duration, while the RTX 5090 achieves nearly identical convergence behavior to the far more expensive V100. Interestingly, V100, despite its high price, exhibits longer convergence time, an effect likely driven by architectural and memory-bandwidth bottlenecks under medium batch sizes and suboptimal tensor-core utilization relative to newer architectures.

Although the H100 provides the fastest convergence, its substantially higher runtime cost may outweigh its time advantage in budget sensitive deployments. Even though the T4 is the cheapest GPU, CHRONOS shows that slightly higher-tier options, such as the A40, can achieve superior cost to convergence efficiency, providing practical feedback for users seeking balanced speed to cost trade offs.

## 5 Discussion

Our evaluation is conducted under controlled training recipes to isolate the relationship between initialization-time signals, model structure, hardware characteristics, and convergence behavior. Specifically, the current CHRONOS feature space includes optimizer type, learning rate, batch size, numerical precision, dataset size, model-level computational descriptors, and hardware descriptors. The convergence estimator uses the model, dataset, and optimization descriptors to estimate the convergence horizon, while the iteration-time estimator uses computational and hardware descriptors to estimate per-iteration runtime.

However, convergence can also be affected by additional recipe-level choices that are not explicitly modeled in the current implementation, including data augmentation strength, label noise, learning-rate schedules, warmup policies, momentum coefficients, and weight-decay values. For this reason, the results should be interpreted as evaluating zero-shot convergence prediction under controlled and reproducible training settings, rather than under arbitrary training recipes.

Extending CHRONOS to include richer recipe descriptors is a natural next step. For example, augmentation strength could be encoded through transformation parameters or policy identifiers, label noise through an

estimated or specified corruption rate, and learning-rate schedules through schedule type, warmup length, decay factor, and final learning rate. We expect these additions to improve generalization in settings where convergence is strongly shaped by recipe-level interventions beyond the architecture and optimizer configuration.

## 6 Conclusion

This work introduced CHRONOS, a unified framework for predicting end-to-end training time and convergence of deep neural networks across heterogeneous hardware. CHRONOS provides zero-shot estimates of total training duration and cost by integrating computational features with initialization-based probes such as gradient norms, NTK traces, and initial loss values. Across a broad spectrum of architectures, from lightweight models like MobileNetV2 and DeiT-Tiny to larger networks, such as ViT, ResNet-50, and DenseNet-121, CHRONOS achieves an average prediction error of 13.7% MAPE for iteration-level execution time and 22.1% MAPE for convergence estimation, enabling accurate and cost-aware forecasts of full training performance.

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
