# OpenReview forum: "Chronos: A Unified Framework for Predicting Training Time and Convergence in Deep Learning"
_TMLR — Under review for TMLR_

### Review · Reviewer_ABEJ · 2026-07-03

**Summary Of Contributions:**

The article presents Chronos as a unified, end-to-end framework for predicting the total time, financial cost, and number of epochs until convergence of deep neural networks before the training completes.

The experimental results indicate that the framework exhibits consistent performance, with low error rates in predicting time per iteration and estimating convergence, and shows improvements over previous work, particularly compared to PreNeT. The work also demonstrates the ability to generalize to different architectures, such as CNNs, Transformers, and MLPs, and to different hardware configurations, including scenarios with GPUs not observed during training.

**Audience:**

Yes

**Audience Explanation:**

The conclusions presented in the article are potentially of interest to a significant portion of the TMLR audience, especially researchers and professionals working in applied machine learning and ML systems. Predicting training cost and time is a recurring problem that directly impacts experimental design and the use of computational resources.

There is also potential interest among researchers in meta-learning and optimization, particularly in using initialization signals to predict training behavior.

**Broader Impact Concerns:**

The article does not have a section dedicated to broader impact considerations. However, given the nature of the proposal, no relevant direct ethical concerns are identified.

Chronos is characterized as an infrastructural tool, aimed at predicting the time, cost, and training behavior, with a focus on computational efficiency. Its use does not directly involve sensitive data or the generation of content.

However, we highly recommend that a Broader Impact Concern be included regarding the risks of an incorrect prediction that generates unexpected costs (time or financial) for researchers who have used Chronos for a prediction that may prove inaccurate during training.

**Claims And Evidence:**

Yes

**Claims Explanation:**

In general, the claims presented in the article are supported by consistent experimental evidence, with a clear methodological description and the use of metrics widely consolidated in the literature, such as MAPE and RMSE. The results are presented quantitatively and comparatively, enabling an objective evaluation of the proposed method's performance.
The experiments indicate that Chronos outperforms previous approaches, especially PreNeT, with significant error reductions, including improvements of 27%-60% in RMSE, depending on the evaluated architecture. In addition, relatively low average error values ​​are reported, such as 13.7% for MAPE for time per iteration and 22.1% for convergence estimation, which reinforces the consistency of the empirical evidence.

The work also demonstrates attention to reproducibility aspects, making the code available in a public repository.

Despite the overall solidity, there are points that partially limit the clarity and verifiability of the evidence. In subsection 4.3, the mention of an ablation study associated with the experimental protocol or the choice of regression model is not sufficiently clear, and this material is not easily located in the repository provided. Furthermore, the convergence metric (MAPE of 22.1%) is not accompanied by a detailed table per model, unlike the time results, which hinders a more in-depth analysis.

**Requested Changes:**

A first point concerns the comparison with related works. Although the article directly compares Chronos with PreNeT, the mention of the Habitat framework (Geoffrey et al., 2021) as related work is not accompanied by a direct experimental evaluation. It is recommended to include this comparison or explicitly justify its absence.

There is a need to clarify and make the ablation study mentioned in subsection 4.3 accessible. The current description is ambiguous regarding its objective, and it is unclear whether it refers to the experimental protocol or the choice of regression model. Furthermore, the material is not readily available in the provided repository. It is recommended to include this study as an appendix to the article, highlighting its main results.

Regarding the convergence estimate, although the article highlights, in item III of the Introduction, an average MAPE of 22.1% for the convergence estimate, a detailed breakdown by model or architecture is lacking. The inclusion of the MAPE metric by architecture, as is done for RMSE in Table 3, is important for evaluating the method's robustness across different architectures.

Several points related to reproducibility could be explained more clearly. The article states in its main contributions (p. 3) and in Section 3.4 (p. 9) that the method uses the NTK trace as one of the signals collected in the initial probing stage ("single batch signals such as gradient norm, NTK trace, and initial loss"), but it is not clear how this metric was calculated in this study.

The term "zero-shot" can be ambiguous, as the method requires processing a real mini-batch for probing (Section 3.4). It is understood that the term is used in the sense of a zero update (since there is no change in the model weights). However, since this probing requires both forward and backward passes, it incurs computational cost. Given this, it is suggested that this terminology be clarified in the text. In addition, it would be highly beneficial to include a discussion of the impact of mini-batch size: what is the suggested size for the survey, and how does the accuracy of the convergence prediction scale with variations in this size?

Finally, we note that the meta-dataset used is predominantly composed of image classification tasks, such as CIFAR, STL-10, and TinyImageNet. This bias may limit the method's generalization to other domains, and a clearer explanation would contribute to a more accurate assessment of the work's scope. Therefore, we suggest that this limitation be included in Section 5 (Discussion) along with the other limitations already explained.

---

### Review · Reviewer_e2x8 · 2026-07-21

**Summary Of Contributions:**

The paper seems to introduce Chronos, a framework that predicts the total training time of deep neural network before fully training it. The work aims to get both iteration time and convergence into a single framework. To this end, Chronos Predictive Engine integrates two modules: the iteration time estimator for the cost of one training iteration, and the convergence estimator for the number of epochs required for the model convergence. Then, Total Training Time = Predicted Iteration Time × Iterations per Epoch × Predicted Convergence Epochs.

Iteration time estimator seems to use analytical model features such as FLOPs, parameter size, ... and uses XGBoost to predict iteration runtime across GPUs. Convergence estimator uses configuration features together with a single initialization probe consisting of one forward and backward pass to extract gradient norm, NTK trace proxy, and initial loss. This is used to predict the number of epochs until convergence. This is evaluated on CNNs, MLP-mixers, ViT on several NVIDIA GPUs.

**Audience:**

No

**Audience Explanation:**

The paper seems to introduce a unified prediction framework which seems interesting. In fact, the paper only requires one-epoch run along with static configurations to estimate the cost and seems to argue that it works on various hardware. This seems quite interesting. However, it is difficult to understand whether the paper presents substantial advances against prior arts in each domain (iteration time prediction and convergence prediction). It seems quite straightforward to think of using best from each domain then multiplying to get the prediction with equation: Total Training Time = Predicted Iteration Time × Iterations per Epoch × Predicted Convergence Epochs.

**Claims And Evidence:**

No

**Claims Explanation:**

While iteration time estimator using the suggested features seems somewhat reasonable, there is a fundamental question about how accurate it can be. For example, depending on schedule, layer type, dynamic behaviors, the iteration time could be way off. It would be great to perform a stronger evaluation in these respect. In fact, LOMO evaluation performed in the paper does not seem to be the most effective way of testing. In this paper, all held-out models seem to be vision networks trained on similar image-classification datasets. A stronger test of generalization would include entirely different model families.

Also, I am not sure how much theoretical understanding of the overall "optimization theory" has went into using initialization statistics to predict convergence across diverse architectures and optimization settings. It would be great to have more details in this regard.

It also seems to be missing comparisons to the prior arts of each domain (iteration time prediction and convergence prediction). Instead of simply comparing against

**Requested Changes:**

Critical:
* Please provide evaluations of iteration time on more variegated set of architectures and models. It would be especially better to include more cross-validations. For example, language vs image, image vs tabular, etc. and CNN vs CNNs with more separable convolutions, CNN vs transformers, CNN vs MoEs, etc.
* Please perform an evaluation of Chronos estimators compared to the prior state-of-the-art approaches in each domain (iteration time prediction and convergence prediction).
* It seems quite straightforward to think of using best from each domain then multiplying to get the prediction with equation: Total Training Time = Predicted Iteration Time × Iterations per Epoch × Predicted Convergence Epochs. Can you compare against this?

Less critical:
* Please provide some theoretical analysis to the estimators (although not critical, it would require significantly more detailed and scaled experiments otherwise). This is especially for the convergence. It would be far more convincing if the work includes more detail on how it relates to the research performed in optimization theory.

---

### Review · Reviewer_EVRg · 2026-07-21

**Summary Of Contributions:**

The paper proposes Chronos, a framework that predicts end-to-end training time and cost of DNNs before full training. It combines two offline-trained regressors: an iteration-time estimator using analytical compute/memory features plus GPU descriptors, and a convergence estimator using static descriptors plus a single-mini-batch initialization probe. Multiplying epochs-to-convergence, iterations-per-epoch, and per-iteration time yields total duration and monetary cost.

The motivation is compelling. The paper clearly identifies a practical issue: users often start expensive GPU training runs without knowing how long they will take or how much they will cost. Existing methods are limited because benchmarks cover only common models, profiling requires access to the target hardware, and learned predictors usually estimate only iteration time rather than total convergence time. The early signal probe is highly efficient, requiring only one forward and backward pass on a single mini-batch, without an optimizer update. Since this costs less than 0.3% of one training epoch, the method's zero-shot prediction seems practical.

However, the reviewer found issues with the evaluation methodology (Section 4) and with the scope of the claims relative to the evidence, as detailed below.

**Additional Comments:**

No additional comments.

**Audience:**

Yes

**Audience Explanation:**

The reviewer still liked the idea presented in this manuscript. Predicting end-to-end training time and cost before committing GPU resources is a practically important problem, and the results suggest that combining cheap initialization-time probes with analytical runtime features is a practical approach that would interest researchers planning experiments under hardware and budget constraints.

**Broader Impact Concerns:**

No concerns.

**Claims And Evidence:**

No

**Claims Explanation:**

Novelty is weakened by Zancato et al. [1]. The paper's main idea — predicting training duration before training using initialization-time NTK/gradient signals — was already established by Zancato et al. (2020). Although this paper uses a different mechanism, which is a learned meta-level regression instead of solving linearized training dynamics, the high-level idea remains very similar. The contribution may therefore appear to be an engineering combination of existing techniques rather than a fundamentally new prediction mechanism.

The random row-wise split used in Table 3 can place configurations sharing the same model–dataset–GPU triple in both training and test sets, differing only in hyperparameters such as batch size or learning rate. Consequently, the reported 27–60% improvement over PreNeT may primarily reflect closely related configurations only rather than generalization to unseen models, datasets, or hardware. The head-to-head comparison against PreNeT is only performed under this split, whereas the grouped evaluation protocols report Chronos alone.

The model-agnostic claim is evaluated only for from-scratch training of vision architectures. Modern application is dominated by fine-tuning and parameter-efficient methods such as LoRA, where most parameters are frozen. In these settings, initialization-time loss, gradient, and NTK features can differ substantially from those observed in the training meta-dataset, and analytical features such as optimizer-state bytes and the backward-pass multiplier κ would misrepresent the actual workload.

[1] Zancato, Luca, et al. "Predicting training time without training." Advances in Neural Information Processing Systems 33 (2020): 6136–6146.

**Requested Changes:**

Report the comparison against PreNeT under the grouped evaluation protocols (LOMO, LODO, and leave-one-GPU-out), not only under the random split. Please also explain the inconsistency between the 80/10/10 split described in Section 4.3 and the 80/20 train-test split stated in the Table 3 caption. (The reviewer is unsure whether this is too trivial to pinpoint.)
Add baselines for the convergence estimator: an adapted version of Zancato et al. (2020).
Evaluate Chronos on fine-tuning and LoRA.